

# Species recovery and recolonization of past habitats: lessons for science and conservation from sea otters in estuaries

Brent B. Hughes[1,2], Kerstin Wasson[3,4], M. Tim Tinker[4,5], Susan L. Williams[6], Lilian P. Carswell[7], Katharyn E. Boyer[8], Michael W. Beck[9], Ron Eby[3], Robert Scoles[3], Michelle Staedler[10], Sarah Espinosa[4], Margot Hessing-Lewis[11], Erin U. Foster[11,12], Kathryn M. Beheshti[4], Tracy M. Grimes[13], Benjamin H. Becker[14], Lisa Needles[15], Joseph A. Tomoleoni[5], Jane Rudebusch[16], Ellen Hines[16] and Brian R. Silliman[2]

[1] Department of Biology, Sonoma State University, Rohnert Park, CA, USA
[2] Division of Marine Science and Conservation, Nicholas School of the Environment, Duke University, Beaufort, NC, USA
[3] Elkhorn Slough National Estuarine Research Reserve, Watsonville, CA, USA
[4] Department of Ecology and Evolutionary Biology, University of California, Santa Cruz, Santa Cruz, CA, USA
[5] U. S. Geological Survey, Western Ecological Research Center, Santa Cruz, CA, USA
[6] Department of Evolution and Ecology, Bodega Marine Laboratory, University of California, Davis, Bodega Bay, CA, USA
[7] Ventura Fish and Wildlife Office, United States Fish and Wildlife Service, Ventura, CA, USA
[8] Estuary & Ocean Science Center, Department of Biology, San Francisco State University, Tiburon, CA, USA
[9] Institute of Marine Sciences, University of California, Santa Cruz, Santa Cruz, CA, USA
[10] Monterey Bay Aquarium, Monterey, CA, USA
[11] Hakai Institute, Heriot Bay, BC, Canada
[12] Applied Conservation Science Lab, University of Victoria, Victoria, BC, USA
[13] Department of Biology, San Diego State University, San Diego, CA, USA
[14] Point Reyes National Seashore, United States National Park Service, Point Reyes Station, CA, USA
[15] Center for Coastal Marine Sciences, Department of Biological Sciences, California Polytechnic State University—San Luis Obispo, San Luis Obispo, CA, USA
[16] Estuary & Ocean Science Center, Department of Geography and Environment, San Francisco State University, Tiburon, CA, USA

Corresponding author
Brent B. Hughes,
hughes@sonoma.edu

## ABSTRACT

Recovering species are often limited to much smaller areas than they historically occupied. Conservation planning for the recovering species is often based on this limited range, which may simply be an artifact of where the surviving population persisted. Southern sea otters (*Enhydra lutris nereis*) were hunted nearly to extinction but recovered from a small remnant population on a remote stretch of the California outer coast, where most of their recovery has occurred. However, studies of recently-recolonized estuaries have revealed that estuaries can provide southern sea otters with high quality habitats featuring shallow waters, high production and ample food, limited predators, and protected haul-out opportunities. Moreover, sea otters can have strong effects on estuarine ecosystems, fostering seagrass resilience through their consumption of invertebrate prey. Using a combination of literature reviews, population modeling, and prey surveys we explored the former estuarine habitats outside the current southern sea otter range to determine if these estuarine

habitats can support healthy sea otter populations. We found the majority of studies and conservation efforts have focused on populations in exposed, rocky coastal habitats. Yet historical evidence indicates that sea otters were also formerly ubiquitous in estuaries. Our habitat-specific population growth model for California's largest estuary—San Francisco Bay—determined that it alone can support about 6,600 sea otters, more than double the 2018 California population. Prey surveys in estuaries currently with (Elkhorn Slough and Morro Bay) and without (San Francisco Bay and Drakes Estero) sea otters indicated that the availability of prey, especially crabs, is sufficient to support healthy sea otter populations. Combining historical evidence with our results, we show that conservation practitioners could consider former estuarine habitats as targets for sea otter and ecosystem restoration. This study reveals the importance of understanding how recovering species interact with all the ecosystems they historically occupied, both for improved conservation of the recovering species and for successful restoration of ecosystem functions and processes.

## INTRODUCTION

Widespread declines of top predator populations have profoundly altered ecological relationships, resulting in the degradation or collapse of many ecosystems (*Jackson et al., 2001*; *Estes et al., 2011*; *Ripple et al., 2014*). Because many of these losses occurred well before systematic ecological studies could document their effects, our understanding of ecological baselines has been fundamentally skewed (*Jackson et al., 2001*). Sea otters (*Enhydra lutris*) provide an excellent example of a top predator whose decline has dramatically altered ecological relationships. Coastal habitats along the North Pacific Rim between Hokkaido, Japan and central Baja California, Mexico formerly supported large numbers of sea otters, but the fur trade of the eighteenth and nineteenth centuries reduced sea otter numbers worldwide from an estimated 150,000–300,000 pre-exploitation (*Kenyon, 1969*; *Johnson, 1982*) to approximately 2,000 post-exploitation (*Kenyon, 1969*). Northern sea otter (*E. lutris kenyoni*) populations around Washington, British Columbia and Southeast Alaska have benefitted from successful reintroductions, whereas in other places natural range expansion has ceased (i.e., southern sea otters—*E. lutris nereis*—in California), or the population is in a state of decline, such as in southwest Alaska (*Bodkin, 2015*). As sea otters have recovered from near extinction over the past decades, they have been the subject of intensive research aimed at furthering their recovery and understanding more fully their role as keystone predators in coastal ecosystems.

While sea otters are closely associated with kelp forests, historical and paleoecological evidence makes clear that southern sea otters were once abundant in California estuaries. Archaeological sites from central California estuaries suggest that Native Americans hunted sea otters, as their bones have often been found in middens throughout

San Francisco Bay, Drakes Estero, Elkhorn Slough, and Morro Bay, representing a period spanning most of the last 7,000 years (*Broughton, 1999*; *Jones et al., 2011*, Pt. Reyes National Seashore Museum Collections) (Fig. 1A). In the largest of these middens in the San Francisco Bay area (Fig. 2A), the Emeryville Shellmound, sea otter skeletal remains were the most abundant among a wide variety of other mammalian skeletal remains (*Schenck, 1926*). *Schenck (1926)* noted that sea otters "occurred at all depths" of the shellmound, and that they were "of prime importance in the economy of the inhabitants."

Early accounts by Spanish explorers documented sea otters' widespread use of San Francisco Bay, ranging from San Jose in the south to Richardson Bay in the north (Fig. 2A). *Ogden (1941)* notes, "San Francisco Bay abounded in otters. Apparently, they not only swam around in the bay but frequented the numerous estuaries and even hauled up on the shore." Sea otters probably numbered in the thousands in this estuary prior to being driven to local extinction by over-hunting (*Schenck, 1926*). The Aleut (Alaskan Native Americans), who were much more skilled otter hunters than the Ohlone (California Native Americans), were forced to join European hunting expeditions and were capable of killing and retrieving hundreds of sea otters daily from San Francisco Bay (*Ogden, 1941*).

The southern sea otter was widely believed to be extinct until a remnant population of fewer than 50 individuals was reported in 1914 near Pt. Sur along the rugged Big Sur coast in California (*Bryant, 1915*). As of 2018, this population had grown to over 3,000 individuals and expanded its geographic range to include more than 400 km of coastline (*Tinker & Hatfield, 2017b*). Southern sea otter numbers in 2018 reached the criterion established by the *U.S. Fish and Wildlife Service (2003)* for delisting consideration under the Endangered Species Act, but they are far below the estimated carrying capacity of approximately 16,000 for California (*Laidre, Jameson & DeMaster, 2001*) and remain restricted to a fraction of their historic range (*U.S. Fish and Wildlife Service, 2003*; *Lafferty & Tinker, 2014*). Because the initial recovering population occurred in kelp forests along the open coast throughout the sea otter range, foundational studies on the ecology of sea otters have focused on this habitat (*Lowry & Pearse, 1973*; *Estes & Palmisano, 1974*; *Simenstad, Estes & Kenyon, 1978*). A similar of baseline knowledge on alternate habitat types has been seen for other top predators around the world, which rather than occupying seemingly unique habitats for the first time, are in fact reoccupying historical habitat (*Silliman et al., 2018*). As a result of historical accident (i.e., the location of the surviving population), sea otters in California have become closely associated with kelp forests in the minds of both the public and scientists. However, southern sea otters historically occupied estuaries as well as the open coast (*Ogden, 1941*).

Starting in the 1970s, descendants of the Big Sur sea otters began to appear in central California estuaries (Fig. 1A; *Lubina & Levin, 1988*), and by the 1990s, they were common in one estuary, Elkhorn Slough (*Feinholz, 1998*). The current southern sea otter recovery plan does not include estuaries as target habitats (*U.S. Fish and Wildlife Service, 2003*), although estuaries may represent a key component of continued southern sea otter recovery, and this has been recognized by recent status reports (*U.S. Fish and Wildlife Service, 2015*). Several benefits of estuarine habitats for sea otters have been

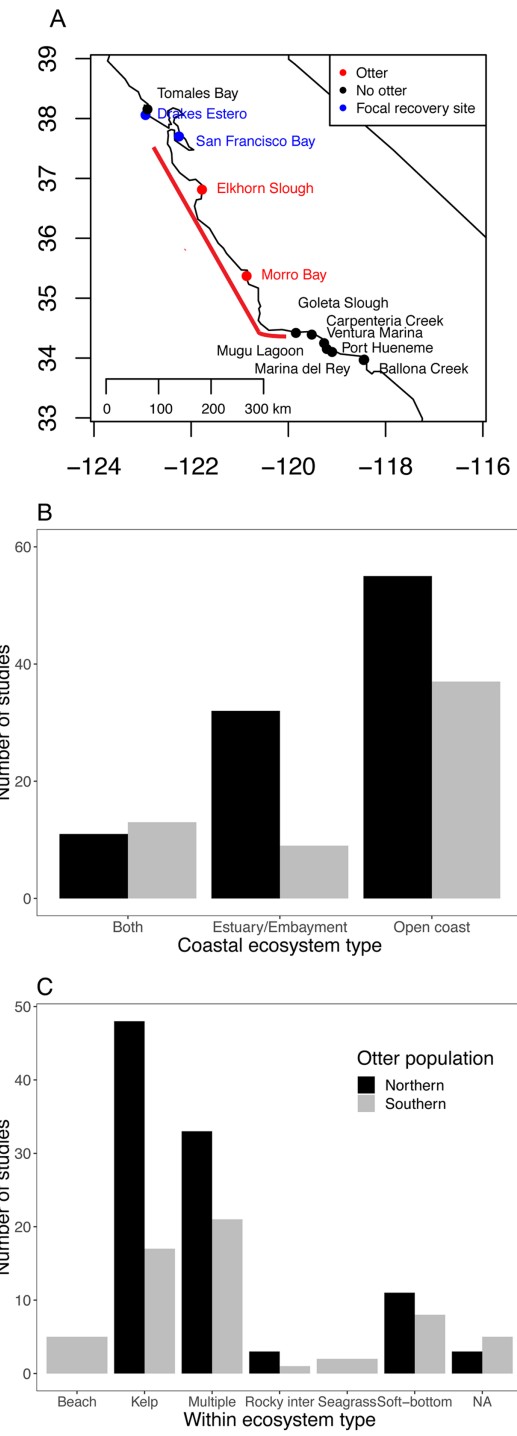

**Figure 1 Current status of sea otter distribution and research effort in different ecosystems and habitats.** (A) Range of southern sea otters in California estuaries as of 2018. Red points indicate estuaries where permanent populations currently exist. Blue and black points indicate estuaries within 100 km of the current range (indicated by the red line) (*Tinker & Hatfield, 2017b*), and blue and red sites are focal sites used in this study. (B and C) Literature search documenting ecosystem habitat use by sea otters for: (B) three coastal ecosystems, and (C) habitats within the three ecosystems. The Southern population (*Enhydra lutris nereis*) represents California (south of 40°N), and the Northern population covers sea otters (*E. lutris kenyoni*) north of 40°N.

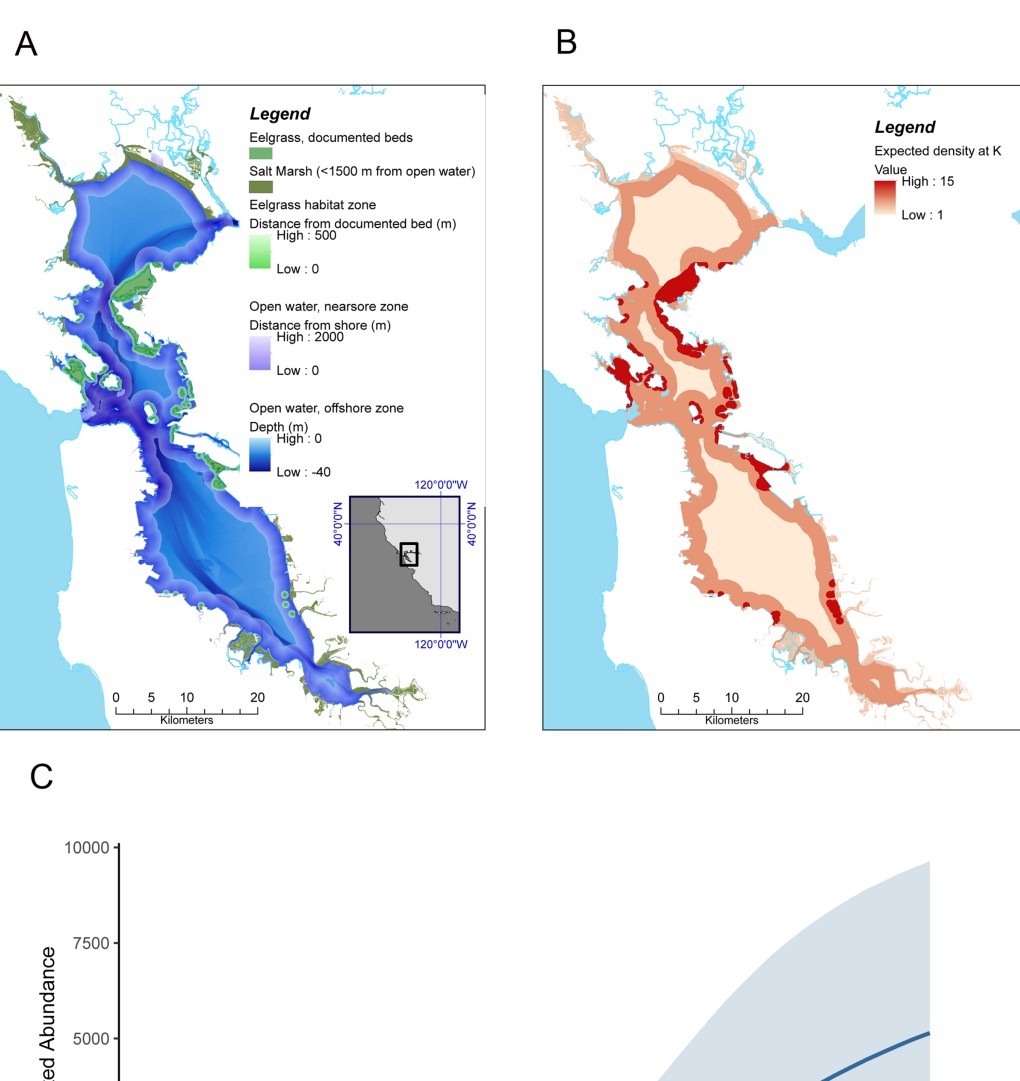

**Figure 2 Sea otter habitats (A) and growth model for San Francisco Bay (B and C).** (A) Estuarine habitats in San Francisco Bay that correspond to habitats currently used by sea otters in Elkhorn Slough, including saltmarsh with tidal creeks, open water, and eelgrass (*Zostera marina*). (B) Expected distribution and projected density at *K* of sea otters in San Francisco Bay, computed by applying the lower 95% CI of estimated carrying capacity for the equivalent habitats in Elkhorn Slough (these densities correspond to approximately 38% of the current Elkhorn Slough density); (C) Projected growth of a resident sea otter population in San Francisco Bay over a 50 year period, initialized with 20 animals at year 1 and assuming similar dynamics as observed in Elkhorn Slough (but using conservative estimates of carrying capacity). Grey band encompasses 90% of simulations.

documented for Elkhorn Slough, such as low predation risk (i.e., lack of predators like white sharks). Along the northern and southern limits of the current range, non-consumptive bites by white sharks (*Carcharodon carcharias*) are the greatest cause of mortality and a primary factor limiting continued expansion into adjacent areas with a higher per-capita abundance of prey (*Tinker et al., 2016*).

Other benefits of estuaries for sea otters include ample prey (*Hughes et al., 2013*), haul out opportunities (*Eby et al., 2017*), and sheltered nursery habitat. Sea otters in Elkhorn Slough haul out extensively on salt marshes, even during daylight, and spend much more time hauled out than do sea otters that have been studied along the open coast (*Eby et al., 2017*). Hauling out may be more common in estuaries due to the availability of easily accessible, low-relief habitat, lower disturbance by humans than on coastal beaches, and high densities of sea otters, allowing for group vigilance to potential predators (*Eby et al., 2017*). Hauling out can provide important energetic benefits for sea otters, which unlike other marine mammals, have no blubber insulation, an exceptionally high metabolic rate, and exhibit high thermal lability in sea water (*Costa & Kooyman, 1982*). The energetic benefits of hauling out may be especially important for females with pups, as these animals incur the increased energy demands of lactation and pup-rearing (*Thometz et al., 2014*). These benefits will likely be provided by other California estuaries as they are recolonized by sea otters in the future. Moreover, extensive shallow and productive estuarine habitats may allow for accelerated population growth at the regional scale.

Virtually all findings on sea otters' top-down effects on coastal food webs, as well as our understanding of the challenges to their recovery, were developed from open-coast studies (*Estes & Palmisano, 1974*; *Estes et al., 1998*, *2003*). Nevertheless, recent studies (*Hughes et al., 2013*, *2016*; *Hessing-Lewis et al., 2017*) suggest that sea otters can generate a trophic cascade in estuarine ecosystems, albeit via different pathways than on the open coast. Given the relatively recent recolonization of estuarine habitats and new insights from recent and ongoing research (*Hughes et al., 2013*, *2016*; *Hessing-Lewis et al., 2017*), it is timely to synthesize the state of our understanding of the relationship between sea otters and estuaries, past and present, and to consider the southern sea otter's recovery and restoration potential in estuarine habitats within the subspecies' historic range.

Here, we use multiple lines of inquiry to assess the potential role of estuaries for supporting sea otter recovery. First, we explore the current scientific literature (post 1950) to better understand how research effort on sea otters compares in different coastal ecosystems. Next, we use a state-space model to estimate the processes of density-dependent population growth for sea otters in one estuarine system (Elkhorn Slough) where sea otters have fully recovered, and we apply this model to another large estuary (San Francisco Bay) outside the current range of sea otters to evaluate its potential to support sea otters. Finally, we evaluated prey data from estuaries within and immediately outside the current sea otter range to determine if there is adequate prey availability to support sea otter populations in estuaries outside the current sea otter range.

## MATERIALS AND METHODS

### Historical research effort on estuaries as sea otter habitat

We summarized research efforts in different ecosystems using a Web of Science search with the following terms: TOPIC: (sea otter*) AND TOPIC: (marsh* or salt marsh* or kelp* or seagrass* or rocky* or soft-bottom* or nearshore* or coast* or subtidal* or intertidal* or beach* or embayment* or * bay* or shore*). Results were accessed on March 3, 2017. We divided studies by two categories: (1) northern (*E. lutris kenyoni*) and southern (*E. lutris nereis*) sea otter populations, and (2) coastal ecosystem type (open coast, estuary/embayment, and both). We further subdivided population and coastal ecosystem type into habitats within the two ecosystems (beach, kelp, rocky intertidal, seagrass, soft-bottom, and multiple).

### State-space model of sea otter growth and equilibrium abundance in a California Estuary

At the present time Elkhorn Slough is the only estuary in California in which a distinct and self-sustaining population of sea otters has become established (*Kvitek et al., 1988*; *Maldini et al., 2012*; *Hughes et al., 2013*; *Eby et al., 2017*), although with continued range expansion it is likely that other estuaries (including San Francisco Bay) will eventually be colonized given the current population is <40 km away from the mouth of San Francisco Bay (*Tinker & Hatfield, 2017a*). Sea otters initially moved into the Slough in 1984, however the early colonizers were mostly or entirely males (consistent with typical patterns of sea otter range expansion; *Lafferty & Tinker, 2014*). Reproductive females did not become established in Elkhorn Slough until the early 2000s, largely as a result of the addition of rehabilitated juvenile female otters by the Monterey Bay Aquarium (*Mayer et al., 2019*), after which intrinsic growth of the Elkhorn Slough population was rapid (*Estes & Tinker, 2017*; *Mayer et al., 2019*). It is therefore reasonable to characterize the first 15 years of sea otter occupation in Elkhorn Slough as a population-establishment phase, and we use data on population trends from 2000–2018 to assess the potential for growth and equilibrium of established populations in estuarine habitats. Annual survey data for Elkhorn Slough were extracted from U.S. Geological Survey population survey data available in an online repository (*Tinker & Hatfield, 2017b*) and are also provided in Dataset S1.

To estimate key parameters of sea otter population dynamics in estuaries, we fit a Bayesian state space model (SSM) to the time series of survey data for Elkhorn Slough using established methods (*Stroud, Muller & Sanso, 2001*). The rationale of an SSM is as follows: a process model of population dynamics (in this case, a stochastic logistic growth model of density-dependent population growth) is fit to a time series of population counts using Markov Chain Monte Carlo (MCMC) methods. This method enables disentanglement of process error (i.e., environmental stochasticity, the unexplained but real variation in annual growth rate, or $\lambda$) from observer error (i.e., measurement and sampling error, the unexplained year-to-year differences in sightability that do not reflect

true abundance). We assume that true abundance, $N_t$, is a latent variable whose dynamics are described as:

$$N_t = N_{t-1} \cdot \exp\left(r\left[1 - \frac{N_{t-1}}{K}\right] + \varepsilon_t\right), \quad \varepsilon_t \sim N(0, \sigma_p) \tag{1}$$

In Eq. (1), $r$ is the intrinsic rate of population growth, $K$ is the equilibrium abundance, or carrying capacity, and $N(0, \sigma_p)$ represents a random normal variable with mean 0 and standard deviation $\sigma_p$, a fitted parameter representing process error or environmental stochasticity. We note that Eq. (1) reflects simple logistic growth, a special case of theta-logistic growth where the implied value of $\theta$ (a parameter determining the shape of the relationship between $\log(\lambda)$ and density) is set to 1. The assumption of simple logistic growth is recommended for data sets where there are insufficient data to reliably estimate $\theta$ (*Clark et al., 2010*) and is consistent with other simple models of sea otter population growth (*Tinker, 2015*; *Tinker et al., 2019a*).

In addition to obtaining estimates of $r$ and $K$ appropriate for estuarine habitats, we also derive estimates of equilibrium densities specific to habitat types, $k_h^d$. We define three general habitat types for sea otters in Elkhorn Slough: (1) nearshore open water, (2) eelgrass beds, and (3) salt marsh habitats (including both tidal creeks and adjacent pickleweed beds). To calculate $k_h^d$, we define $A_h$ as the area of habitat type $h$ available in the Slough (in units of km$^2$) and $p_h$ as the proportion of all surveyed sea otters occurring in each habitat type $h$. We then calculate the equilibrium densities for each habitat type as:

$$K_h^d = \frac{p_h \cdot K}{A_h} \tag{2}$$

To fit the process model defined by Eq. (1), we relate the latent variable, $N_t$, to the observed survey counts at Elkhorn Slough, $C_t$, by assuming that the counts are distributed as a negative binomial distribution with mean $N_t$ and variance $\sigma_o^2$:

$$C_t \sim NB(a, b), \quad a = \frac{N_t^2}{(\sigma_o^2 - N_t)}, \quad b = 1 - \frac{(\sigma_o^2 - N_t)}{\sigma_o^2} \tag{3}$$

where $\sigma_o$ represents observer error, another fitted parameter. Equation (3) implies that survey counts will, on average, be centered around true abundance, and thus does not account for imperfect detection. We recognize this is unrealistic, as some (small) proportion of animals are invariably missed on visual surveys (*Estes & Jameson, 1988*); however, the un-replicated complete count method used to census otters in California does not provide a means of estimating a sightability correction factor. Consequently, for consistency with previous analyses of these survey data (*Lubina & Levin, 1988*; *Tinker et al., 2006*; *Lafferty & Tinker, 2014*), we make the simplifying assumption that counts are centered around true abundance, recognizing that this means that our estimates of $N_t$ are biased low to a small (but unmeasured) degree.

We estimated parameters in Eqs. (1–3) using Bayesian MCMC methods, implemented using JAGS ("Just Another Gibbs Sampler," Code S1) and the R programming language (*Plummer, 2003*; *R Development Core Team, 2015*). We used vague half-Cauchy priors for $K$, $N_1$, and all variance parameters, and an informative normal prior for $r$ with mean

0.2 and standard deviation 0.1, based on the results of numerous studies showing that $r_{max}$ in recently established populations is consistently within the range 0.20–0.25 (*Estes, 1990*; *Gerber et al., 2004*; *Lafferty & Tinker, 2014*; *Tinker, 2015*; *Tinker et al., 2019a*). After model burn-in of 10,000 simulations we saved 500 samples for each of 20 chains, for a total of 10,000 samples used to calculate posterior distributions for all parameters and for the latent variable $N_t$ (Fig. S1). Model convergence was evaluated by graphical examination of the CODA traces, and by ensuring that the R-hat statistic was <1.01 for all parameters. We also conducted posterior predictive checking (PPC; *Gelman et al., 2000*), creating a scatter plot of a discrepancy measure (summed deviance of observed counts) for replicate (simulated) vs actual (observed) data and reporting the associated Bayesian *P*-value (requiring $0.25 < P < 0.75$; *Gelman et al., 2014*). We then summarized the mean, median, and 95% credible intervals (CI) for the posterior distributions of each parameter.

## Applying state space model results to San Francisco Bay

We conservatively took the lower bound of the 95% CI of $k_h^d$ for Elkhorn Slough habitats as our point estimates for equilibrium densities of the equivalent habitats in the San Francisco Bay estuary. A fourth habitat type, offshore open water (defined as areas >2,000 m from shore but still within the 40 m depth zone), occurred in San Francisco Bay but not in Elkhorn Slough. Based on extensive data on the depths of sea otter foraging dives (*Bodkin, Esslinger & Monson, 2004*; *Tinker et al., 2007*; *Thometz et al., 2016*), individual habitat use data from Washington (*Laidre et al., 2009*) and California (*Tinker et al., 2017*), and survey data from large embayments in Alaska such as Prince William Sound, Yakutat Bay and Glacier Bay (*Esslinger & Bodkin, 2009*; *Esslinger et al., 2015*; *Tinker et al., 2019a*), it is clear that sea otters utilize such offshore habitats, but that the densities of animals in offshore areas are substantially lower than densities in the nearshore zone. For example, using published census data for southern sea otters from 1985 to 2018 (*Tinker & Hatfield, 2017b*), and restricting analysis to areas of soft sediment habitat along the open coast, we calculated that the number of otters observed >2,000 m from shore was just 15% of the number counted ≤2,000 m from shore. To obtain an estimate of equilibrium density for offshore areas <40 m depth in San Francisco Bay ($k_4^d$), we multiplied the equilibrium density for nearshore open water ($k_1^d$) by a correction factor, $\Phi = 0.1$, thus conservatively assuming that offshore densities would be ~10% of nearshore densities. We used multiple GIS sources to characterize the four different sea otter habitats in San Francisco Bay, including EcoAtlas (ecoatlas.org), the California Aquatic Resources Inventory for wetland habitats, and the Eelgrass Aquatic Resources for eelgrass habitats.

We applied our conservative estimates of habitat-specific equilibrium densities derived from Elkhorn Slough to GIS layers of habitat cover in San Francisco Bay, then summed across habitats to obtain a total estimated abundance at $K$ for San Francisco Bay (Table 1). We used this value of $K$, together with parameter values for $r$ and of $\sigma_p$ estimated from Elkhorn Slough, to parameterize 500,000 forward simulations of population growth in San Francisco Bay (achieved by iteratively solving Eq. (1)). We drew all parameter values from their respective posterior distributions to account for parameter uncertainty. For all simulations we initialized the population ($N_1$) with 20 sea

**Table 1 Summary statistics from state-space model of sea otter population dynamics in Elkhorn Slough.** (A) Fitted parameters include process error ($\sigma_p$), observer error (expressed as a CV), intrinsic growth rate ($r$) and carrying capacity of Elkhorn Slough ($K$). Bayesian diagnostic statistics include Monte Carlo error and "R-hat." (B) Summary statistics for derived parameters, including estimated density at $K$ for Elkhorn Slough, lower 95% CI for density at $K$, projected habitat-specific equilibrium densities ($k_h^d$, based on lower 95% CI for estimated $K$), and projected $K$ values for San Francisco Bay (assuming offshore open water densities are 10% of nearshore open water densities).

**(A)**

| Model Parameter | Mean | SD | Lower95 | Median | Upper95 | MCerr | R-hat |
|---|---|---|---|---|---|---|---|
| $\sigma_p$ | 0.31 | 0.08 | 0.16 | 0.3 | 0.48 | 0.002 | 1.003 |
| $CV_O$ | 0.1 | 0.08 | 0 | 0.09 | 0.26 | 0.001 | 1.002 |
| $r$ | 0.22 | 0.07 | 0.1 | 0.21 | 0.34 | 0.001 | 1.001 |
| $K$ | 160.37 | 88.09 | 49.45 | 132.65 | 367.45 | 0.893 | 1.001 |

**(B)**

| Derived parameters: equilibrium densities for Otters $\times$ km$^{-2}$ (Elkhorn Slough) | | $K$ (San Francisco Bay) | |
|---|---|---|---|
| $K_{mean}^d$ | 27.1 | | |
| $K_{lowerCI}^d$ | 8.4 | | |
| $K_{nearshore}^d$ | 8.9 | $K_{nearshore}$ | 4,228 |
| $K_{eelgrass}^d$ | 19.2 | $K_{eelgrass}$ | 1,598 |
| $K_{saltmarsh}^d$ | 4.5 | $K_{saltmarsh}$ | 425 |
| $K_{offshore}^d$ | 0.9 | $K_{offshore}$ | 356 |
| | | Total | 6,607 |

otters: assuming a female:male ratio of 2:1, this corresponds (approximately) to the 14 rehabilitated juvenile female sea otters successfully introduced to Elkhorn Slough between 2002 and 2014, which contributed the majority of population growth during this period (*Mayer et al., 2019*). We also assumed there would be a population establishment phase similar to that seen for Elkhorn Slough, and thus forced the mean growth rate to 0 for the first 15 years. We summarized results of these simulations by plotting the mean expected trajectory and 90% CI for 50 years of population dynamics. Finally, we examined the sensitivity of the simulation results to our various assumptions and input parameters. Specifically, we sequentially reduced each model parameter ($k_h^d$, $\Phi$, $r$ and $\sigma_p$) by 10% of its default value, while holding other parameters constant, and measured the proportional response in $N'_{50}$, the mean projected abundance of the population after 50 years.

## Prey availability in estuaries outside the sea otter range

To evaluate prey availability at sites within (Elkhorn Slough and Morro Bay) and outside (San Francisco Bay and Drakes Estero; Fig. 1A) the current southern sea otter range, we took two approaches: first, we assembled existing prey species lists (excluding Drakes Estero); and second, we sampled each estuary for crab, which have been documented to be important prey for successful sea otter recolonization in estuaries (*Garshelis & Garshelis, 1984*; *Hughes et al., 2013*). To generate sea otter prey species lists we assembled data

from various monitoring programs (see Dataset S3 for references) that have sampled benthic organisms in each estuary.

To sample crab at each estuary we used standardized trapping techniques during summer 2016 (Elkhorn Slough), fall 2016 (Drakes Estero, San Francisco Bay) and winter 2017 (Morro Bay, California Department of Fish and Wildlife, Permit SC-6563). Due to logistical constraints we were unable to sample all four sites within the same season. We used shrimp pots (http://www.westmarine.com—Willapa Shrimp Pots) with modified openings that allow access to large crabs yet still prevent access or entrapment of any larger predators, such as sea otters, birds, and harbor seals. The dimensions of the shrimp pots were 61 × 61 × 23 cm, with four openings that were rigidly supported and ~20 cm in diameter. Each site had seven to eight locations sampled over a single tidal cycle in *Zostera marina* seagrass beds, with the exception of Elkhorn Slough, which had four locations that were sampled multiple times ($n$ = 8–12 times during summer 2016). Each site received one trap per sampling effort, and traps were soaked for ~24 h. Each crab was identified and measured for carapace width, which was converted to edible biomass using power functions from *Oftedal et al. (2007)*. San Francisco Bay crab catch per unit effort (CPUE) included the invasive European green crab (*Carcinus maenas*), caught in an April 2017 crab survey, which we have observed to be a prey item of sea otters in Elkhorn Slough (B. Hughes, 2015, personal observation), but was not included in the biomass conversion estimates in *Oftedal et al. (2007)*. We applied a conservative biomass estimate for green crab using the biomass conversion for *Metacarcinus magister*. We consider this estimate to be conservative since green crab have a relatively narrow carapace width relative to biomass compared to other native Cancrid crab species.

To determine differences in crab prey availability in estuaries within and immediately outside the southern sea otter range we compared differences in edible crab biomass and carapace width of two common estuarine crab prey species: *Romaleon antennarium* and *Cancer productus*. For one site, Elkhorn Slough, we used a historic dataset that surveyed crab in the 1970s (1971–1976, see *Hughes et al., 2013* for a description of the data), to compare to the contemporary crab trapping efforts at Elkhorn Slough and the other three estuaries. The 1970s Elkhorn Slough crab dataset characterized Elkhorn Slough crab populations prior to sea otter recolonization in the early 1980s, and thus helped support our interpretation of differences in crab populations for our 2017 surveys.

Differences in sea otter presence (fixed factor) and site (random factor) of CPUE and size were determined using linear mixed models with the *nlme* package in R v. 3.4.3. A Tukey multiple comparison test was used to compare differences among sites with and without sea otters using the *multcomp* package.

## RESULTS

### Historical research effort on estuaries as sea otter habitat

While many studies continue to be published on outer coast sea otters, our Web of Science search resulted in few investigations examining estuarine sea otters (Fig. 1B; Dataset S1; Code S2), and most of the estuarine studies were based on northern sea otters (*E. lutris kenyoni*). We found only nine studies that focused specifically on estuarine ecosystems

for southern sea otters. Of all the studies we examined ($n$ = 158), only two focused on seagrass (*Z. marina*) habitats (Fig. 1C), and only one to our knowledge has focused on salt marshes (*Eby et al., 2017*). Whereas, kelp was the most dominant single habitat studied for both northern and southern sea otter populations.

## State-space model of sea otter growth and equilibrium

The state-space model fit to survey data from Elkhorn Slough (Dataset S2; Code S3) converged well and provided excellent fit to the data (Fig. S1). Trace plots of posterior samples showed well-mixed chains (Figs. S2 and S3), with all R-hat statistics <1.01 (Table 1), and the PPC scatterplot showed excellent agreement between replicate and observed data sets (Bayesian-$P$ = 0.55; Fig. S4). The estimated instantaneous rate of growth at low density ($r$) was 0.22 (CI$_{95}$ = 0.10–0.33), while the estimated magnitude of environmental stochasticity ($\sigma_p$) was 0.30 (CI$_{95}$ = 0.16–0.47). The estimated value of $K$ for Elkhorn Slough was 160 (CI$_{95}$ = 49–367), and using the lower bound of the 95% CI for $K$ we estimate habitat-specific equilibrium densities ($k_h^d$) for three habitat types: nearshore open water (8.9 otters $\times$ km$^{-2}$), eelgrass beds (19.2 otters $\times$ km$^{-2}$) and saltmarsh habitats (4.5 otters $\times$ km$^{-2}$; Table 1 and Fig. 2B).

## Applying state space model results to San Francisco Bay

Our habitat-specific population growth model provides a conservative estimate for the equilibrium abundance of sea otters in San Francisco Bay of over 6,600 animals (Table 1) and shows that just 20 colonizers could grow to a population of 5,000 or more within 50 years (Fig. 2C). There is a great deal of uncertainty in growth projections, evident in the wide CI bounds around the mean projected trend in Fig. 2C: this variability reflects both the effects of process error (environmental stochasticity) and parameter uncertainty. To explore the latter source of uncertainty we conducted a sensitivity analysis, the results of which indicated that the parameter having the greatest impact on projected abundance was $r$, the rate of growth at low densities (Fig. S5). The estimate of $k_1^d$ (equilibrium density in nearshore open water) had the next greatest impact on $N_{50}^t$, reflecting the predominance of this habitat type in San Francisco Bay. The magnitude of environmental stochasticity ($\sigma_p$) also had substantial influence, with a decrease in $\sigma_p$ leading to an increase in projected abundance.

We note that our model projections are conservative in two respects: they assume lower equilibrium densities than currently seen in Elkhorn Slough, and estimates of the distribution of a key habitat type, eelgrass beds, are based on conservative estimates of the current distribution and do not include positive feedback loops whereby sea otter recovery could promote further seagrass expansion. Furthermore, we excluded certain estuarine habitats (beaches and subtidal flats), which are also known to be used by sea otters (*Kvitek et al., 1988*), but for which reliable density estimates could not be obtained. Despite these caveats, our model results suggest that San Francisco Bay could eventually support a sizeable southern sea otter population, 160% or more of the range-wide population as of 2018.

**Table 2  List of sea otter prey items found in San Francisco Bay.**

| Class | Scientific name | Common name | Reference |
| --- | --- | --- | --- |
| Asteroidea | *Pisaster ochraceus* | Seastar | *USGS Western Ecological Research Center (2019)* |
| Bivalvia | *Clinocarduim nuttallii* | Heart cockle | *USGS Western Ecological Research Center (2019)* |
| Bivalvia | *Macoma* spp. | Clam | *USGS Western Ecological Research Center (2019)* |
| Bivalvia | *Modiolus* | Horse mussel | *USGS Western Ecological Research Center (2019)* |
| Bivalvia | *Mya arenaria* | Soft shelled clam | *Emmett et al. (1991)* |
| Bivalvia | *Mytilus edulis* | Bay mussel | *Emmett et al. (1991)* |
| Bivalvia | *Mytilus galloprovincialis* | Mussel | *Suchanek et al. (1997)* |
| Bivalvia | *Ostrea lurida* | Olympia oyster | *Boyer et al. (2017)* |
| Bivalvia | *Protothaca* sp. | Littleneck clam | *Emmett et al. (1991)* |
| Bivalvia | *Saxidomus nuttalli* | Washington clam | *Skinner (1962)* |
| Bivalvia | *Solen sicarius* | Razor clam | *USGS Western Ecological Research Center (2019)* |
| Bivalvia | *Tagelus californianus* | Jackknife clam | *Emmett et al. (1991)* |
| Bivalvia | *Tellina* spp. | Clam | *Hunt et al. (2001)* |
| Bivalvia | *Tivela stultorum* | Pismo clam | *Skinner (1962)* |
| Bivalvia | *Tresus nuttallii* | Pacific gaper clam | *Emmett et al. (1991)* |
| Bivalvia | *Zirfaea pilsbryi* | Rough piddock | *USGS Western Ecological Research Center (2019)* |
| Decapoda | *Cancer productus* | Red rock crab | *USGS Western Ecological Research Center (2019)* |
| Decapoda | *Carcinus maenus* | European green crab | *Boyer et al. (2017)* |
| Decapoda | *Crangon franciscorum* | Bay shrimp | *Boyer et al. (2017)* |
| Decapoda | *Hemigrapsus oregonensis* | Shore crab | *Boyer et al. (2017)* |
| Decapoda | *Metacarcinus gracilis* | Graceful crab | *Boyer et al. (2017)* |
| Decapoda | *Metacarcinus magister* | Dungeness crab | *Boyer et al. (2017)* |
| Decapoda | *Pachygrapsus crassipes* | Striped shore crab | *USGS Western Ecological Research Center (2019)* |
| Decapoda | *Pagurus* sp. | Hermit crab | *USGS Western Ecological Research Center (2019)* |
| Decapoda | *Palaemon macrodactylus* | Oriental shrimp | *Boyer et al. (2017)* |
| Decapoda | *Pugettia producta* | Kelp crab | *Boyer et al. (2017)* |
| Decapoda | *Romaleon antennarium* | Pacific red rock crab | *Boyer et al. (2017)* |
| Decapoda | *Upogebia* sp. | Mud shrimp | *USGS Western Ecological Research Center (2019)* |
| Echinoidea | *Dendraster excentricus* | Sand dollar | *USGS Western Ecological Research Center (2019)* |
| Gastropoda | *Ilyanassa obsoleta* | Eastern mud snail | *USGS Western Ecological Research Center (2019)* |
| Gastropoda | *Nassa fossatus* | Mud snail | *USGS Western Ecological Research Center (2019)* |
| Gastropoda | *Tegula funebralis* | Black turban snail | *USGS Western Ecological Research Center (2019)* |
| Gastropoda | *Urosalpinx cinerea* | Atlantic oyster drill | *Boyer et al. (2017)* |
| Polychaeta | *Nereis* sp. | Pile worm | *USGS Western Ecological Research Center (2019)* |

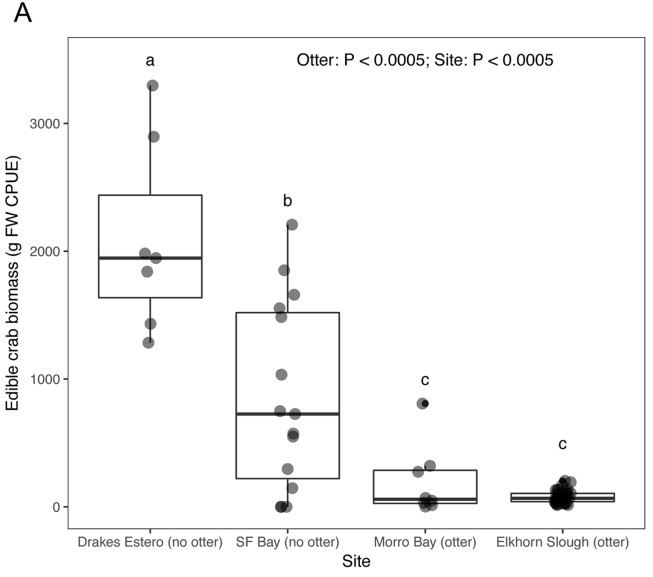

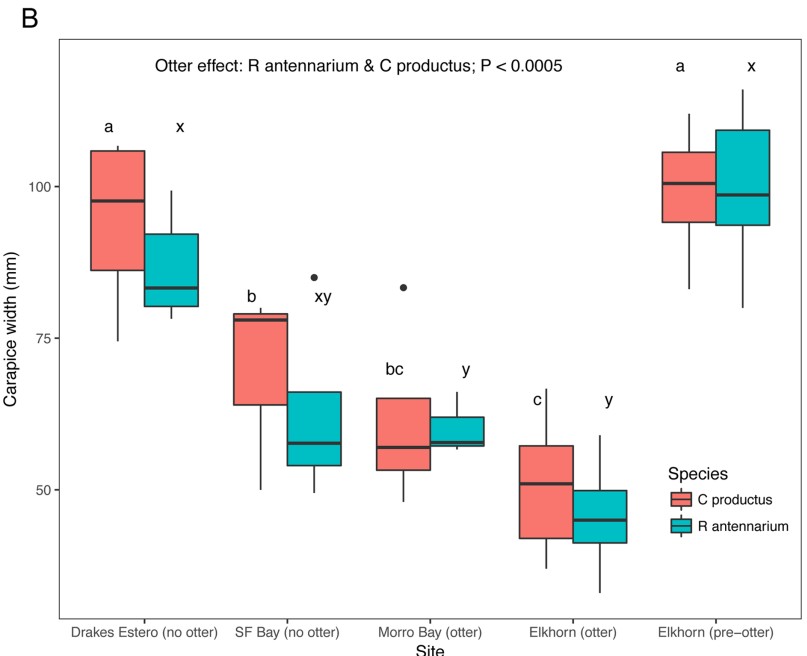

**Figure 3 Crab prey metrics in select California estuaries with and without sea otters.** (A) Catch per unit effort (CPUE) caught in standardized crab traps over a 24 h period, (B) mean carapace width of two common crab prey species (*Romaleon antennarium* and *Cancer productus*) in California estuaries, pre-otter Elkhorn Slough data are from *Hughes et al. (2013)*. Lowercase letters indicate significant differences ($P < 0.05$) between estuaries based on Tukey HSD tests.

## Prey availability in estuaries outside the sea otter range

Our comparison of prey availability in estuaries with sea otters (Elkhorn Slough and Morro Bay) with one of the focal recovery sites (San Francisco Bay) suggests that prey resources in San Francisco Bay are sufficient to sustain a healthy sea otter population. Out of the 52 prey species we identified in the two California estuaries with sea otters, 34 (65%) occurred

in San Francisco Bay (Table 2; Dataset S3). Prey species include some that are of value to commercial fisheries, such as Dungeness crab (*Metacarcinus magister*), littleneck clam (*Leucoma staminea*), and mussels (*Mytilus* spp.), and also include invasive species such as European green crab (*Carcinus maenas*).

Crab survey efforts also indicate that our two focal sea otter recovery sites (San Francisco Bay and Drakes Estero) have an adequate supply of crab prey to support a sea otter population. Both estuaries without sea otters had significantly greater crab biomass caught in standardized traps compared to the estuaries (Elkhorn Slough and Morro Bay) with sea otters (Fig. 3A), and Drakes Estero had a significantly greater crab biomass than San Francisco Bay (Dataset S4; Code S4). When we compared sizes, crabs (*R. antennarium* and *Cancer productus*) from Drakes Estero were significantly larger than crabs from San Francisco Bay (no sea otters) and the two sites with sea otters (Fig. 3B; Dataset S5; Code S4). Furthermore, crabs of both species in Drakes Estero were similar in size to those in Elkhorn Slough prior to sea otter recolonization in the 1970s. Whereas, crab sizes in San Francisco Bay were intermediate between Drakes Estero (no sea otters), Morro Bay (with sea otters), and Elkhorn Slough (before and after sea otter recolonization).

## DISCUSSION

### Paradise lost: sea otters in California estuaries

Despite a substantial increase in population size over the past century, southern sea otters have recolonized only approximately 13% of their historic range (*U.S. Fish and Wildlife Service, 2015*), and long-term population growth has been slow (3–5% per year; *Tinker & Hatfield, 2017a*) compared to recovering populations of northern sea otters (*E. lutris kenyoni*) (*Estes, 1990*), where total population growth rates have been reported to be as high as 17–20% per year depending on location. Although this discrepancy in population growth rates has been interpreted as evidence of persistently high southern sea otter mortality, the observed population trajectory can largely be explained as the combined effect of localized density-dependence and the strong spatial structuring typical of sea otter populations. Adult females, in particular, rarely move more than 15 km per year from their home range center (*Tinker, Doak & Estes, 2008*; *Tarjan & Tinker, 2016*) along the predominantly long, narrow configuration of central California coastal habitat (*Tinker, 2015*). In such elongated and essentially one-dimensional habitats, a large proportion of the population occurs in resource-limited areas, with population growth and range expansion occurring only at the extreme ends of the range (*Bodkin, 2015*; *Tinker, 2015*). In more complex, multi-dimensional habitats (such as those occurring in southeast Alaska and British Columbia) there are typically multiple expansion fronts; therefore, a greater proportion of the population occurs at sub-equilibrium densities, resulting in faster growth rates at the regional scale. Because they afford a diversity of habitats and prey for range expansion, estuaries are among the ecosystems that are expected to be conducive to rapid sea otter population growth.

Two estuaries are located within the current range of the southern sea otter: Elkhorn Slough and Morro Bay. These estuaries are located to the north and south, respectively, of
the central portion of the range, where recovering southern sea otters have resided the longest and are at or near equilibrium densities (*Tinker, Doak & Estes, 2008*; *Thometz et al., 2016*; *Tinker et al., 2019b*). Sea otters recolonized the outer portion of Morro Bay estuary in 1973 (*Lubina & Levin, 1988*) and have recently increased dramatically in number in Morro Bay Harbor (*Tinker & Hatfield, 2017a*), but they have not fully recolonized the inland portion of the estuary. In contrast, sea otters in Elkhorn Slough utilize inland estuarine habitats extensively. Over the past two decades, the Elkhorn Slough population has increased rapidly to more than 100 animals and has only recently shown signs of reaching equilibrium (*Silliman et al., 2018*). The initiation of the period of rapid growth in Elkhorn Slough can partly be attributed to the addition of rehabilitated sea otters from the Monterey Bay Aquarium, specifically to 14 females that survived for >1 year post release and were among the first animals to reproduce in the Slough (*Mayer et al., 2019*). Our initialization of San Francisco simulations with 20 animals was explicitly based on this scenario, although natural colonization would of course be expected to produce similar results.

Although some sea otters still leave Elkhorn Slough occasionally to forage along the outer coast, most appear to be entirely resident within the estuary, using different habitats for feeding, resting and interacting. This recovery demonstrates that estuarine habitats and prey resources can support a high-density, fully resident sea otter population. Estuaries such as Elkhorn Slough are home to diverse and abundant invertebrate communities (*Wasson et al., 2002*), and sea otter prey such as clams, crabs and worms can recruit and grow rapidly (Dataset S3). The high productivity of prey resources has supported decades of high foraging success of the dense sea otter population (*Hughes et al., 2013*; *Silliman et al., 2018*).

The two major estuaries currently inhabited by sea otters (Elkhorn Slough and Morro Bay) comprise 2,400 ha of total habitat and <50 ha of eelgrass habitat as of 2014 (*Hughes et al., 2013*, *2016*; *MBNEP, 2014*). However, if estuaries immediately outside (100 km to the north and south) the current sea otter range are included, the total available estuarine habitat is much greater (Fig. 1A). Considering only those estuaries with eelgrass habitat (*Bernstein et al., 2011*), which could be a catalyst for sea otter colonization and is an indicator of good estuarine water quality and rich invertebrate prey resources, we found 11 more estuaries with a total area of 135,000 ha, which is a 56-fold increase in available estuarine habitat relative to that within the currently occupied range. In comparison, the total available outer coast habitat within the same region is 147,000 ha (*Laidre, Jameson & DeMaster, 2001*). When comparing the coverage of the two dominant habitat-forming species within the sea otter range—eelgrass in estuaries and kelp (e.g., *Macrocystis pyrifera*) on the outer coast—we conservatively estimated there to be 1,750 ha of eelgrass habitat within the projected range, and most of the habitat is on the northern side of the currently occupied range (see *Bernstein et al., 2011*), compared to about 5,700 ha of available kelp habitat along the outer coast (https://earthworks.stanford.edu/catalog/stanford-nf191bc1535). Even using our most conservative estimates, eelgrass alone could provide 23% of the total habitat available for sea otter range expansion over the next few decades within the 100 km projected stretch of mainland California. This

estimate does not take into account the other important estuarine habitats used by sea otters, such as salt marshes, tidal creeks, and mudflats (*Kvitek et al., 1988*; *Eby et al., 2017*), which could greatly add to the total estimate of new sea otter habitat. Taken together, estuaries could provide important recovery habitat for sea otters as they expand to the north and south, and serve as a refuge from sharks and unfavorable outer coast conditions, like El Niño-driven storm and warming events, which reduce the abundance of kelp (*Dayton & Tegner, 1984*; *Edwards, 2004*).

If sea otters recolonize the large estuaries on the California coast, such as San Francisco Bay and Drakes Estero, or if rehabilitated otters were proactively reintroduced, the potential for population growth could be profound (Fig. 2C). As of 2014, eelgrass beds in San Francisco Bay cover about 1,100 ha (*Merkel & Associates, 2014*), and as of 2005 Drakes Estero eelgrass coverage is 300 ha (*Brown & Becker, 2007*). Drakes Estero has a relatively pristine watershed where eelgrass beds have been observed to be stable since a 2005 survey (B Becker, 2019, personal observation). San Francisco Bay's watershed is highly developed with extensive anthropogenic impacts, including wetland conversion and loss (*Gedan & Silliman, 2009*). However, the Bay still supports many of the same habitats and prey resources utilized by southern sea otters in their currently occupied range (Table 2; Fig. 2A).

Sea otters are a prime example of a keystone predator in kelp forests: by regulating herbivorous urchins, sea otters increase the abundance and distribution of kelp (*Estes & Palmisano, 1974*; *Estes et al., 1998*). This well-established paradigm has recently been complemented by the discovery that sea otters play an analogous role in estuaries by enhancing the resilience of seagrass to anthropogenic nutrient inputs (*Hughes et al., 2013*, *2016*). Through their high consumption rates of crabs in Elkhorn Slough, sea otters indirectly benefit seagrasses that are threatened by extreme nutrient loading and eutrophication (*Hughes et al., 2011*, *2013*). By consuming crabs, sea otters can generate a trophic cascade, whereby small grazers (<5 cm) are released from predation by crabs, which enables them to consume shade-causing algal epiphytes that grow on the leaves of eelgrass (*Z. marina*) (*Hughes et al., 2013*, *2016*). Prior to sea otter recolonization of the Elkhorn Slough estuary in 1984, eelgrass was nearly locally extinct. Since the recolonization of sea otters in the last three decades, eelgrass has recovered and expanded in area by >600%, and sea otter population growth positively correlates with eelgrass recovery (*Hughes et al., 2013*). At least in this ecological context, sea otters are capable of mediating the harmful effects of poor water quality on seagrasses without management intervention.

Results showing positive effects of sea otters in estuaries have regional conservation implications, as many estuaries toward the northern and southern limits of the current range of sea otters are highly threatened by anthropogenic stressors (*Greene et al., 2015*; *Hessing-Lewis et al., 2017*). Recently, severe losses of eelgrass to the south of Elkhorn Slough in Morro Bay (>95% since 2007; *MBNEP, 2014*) have been observed, with the exception of areas near the mouth where sea otters permanently reside. A wide range of ongoing human impacts to eelgrass in the north in San Francisco Bay (*Boyer & Wyllie-Echeverria, 2010*) have created the impetus for large-scale restoration.

Eelgrass-facilitating trophic interactions may be one mechanism by which successful conservation and restoration outcomes are maximized.

## An ecosystem-based management approach to decisions about estuarine restoration efforts

The restoration of estuaries has become a priority for land managers because of the shoreline protection, flood management, water filtration, wildlife habitat, recreational and educational opportunities, and other services that estuaries provide. San Francisco Bay, designated an "estuary of national significance" under the Clean Water Act, received $5.2 million in funding through the Environmental Protection Agency in 2017 for water protection and restoration projects in the San Francisco Bay area. Within the San Francisco Bay, the Don Edwards San Francisco Bay National Wildlife Refuge is carrying out the largest wetlands restoration project on the West Coast, the South Bay Salt Pond Restoration project, with the aim of improving the physical, biological, and chemical health of the estuary, including restoring habitat for birds, fish, benthic invertebrates, marine mammals, and plants, some of which are threatened or endangered. Re-occupation by sea otters could assist restoration goals (*San Francisco Bay Subtidal Habitat Goals Report, 2010*) by enhancing the resilience and functioning of estuaries through indirect effects. Potential ecological effects of sea otter recovery extend beyond the top-down impacts of sea otter trophic interactions on seagrass recovery (*Hughes et al., 2013*, *2016*); they also encompass the well-established ecosystem services supported by seagrass systems (*Waycott et al., 2009*), including a growing recognition of seagrass as a sink for atmospheric carbon (*Fourqurean et al., 2012*; *Greiner et al., 2013*). Additionally, sea otter bioengineering effects may have positive feedbacks in soft sediment habitats. Sea otter consumption of clams and subsequent shell deposition can provide hard substrate in an otherwise soft sediment system (*Kvitek & Oliver, 1992*), which could promote the settlement and sub-surface growth of invertebrates and algae, altering community structure and increasing fish habitat. Finally, the physical disturbance created when sea otters forage for invertebrates within seagrass and other soft-bottom ecosystems (*Kvitek et al., 1988*) may have ecosystem feedbacks on benthic communities (*Kvitek & Oliver, 1992*) and ecosystem resiliency.

Based on previously reported rates of range expansion in southern sea otters (*Lubina & Levin, 1988*; *Tinker, Doak & Estes, 2008*; *Lafferty & Tinker, 2014*), sea otter recolonization of other California estuaries would be expected to happen gradually over coming decades. However, as mentioned above, white shark predation now increasingly curtails range expansion at both ends of the range (*Tinker et al., 2016*). In the absence of natural range expansion, an option to be considered is facilitated range expansion, focusing on under-utilized habitats such as estuaries, where shark predation is reduced. This could be accomplished by through the release of sea otters into San Francisco Bay, Drakes Estero or other California estuaries, as was done for Elkhorn Slough (*Mayer et al., 2019*).

The reintroduction of a keystone predator to the estuarine ecosystems it once occupied involves potential trade-offs. The strength of sea otters' ecosystem effects is a result of their high rates of consumption of invertebrate prey. For example, estuaries with sea otters

A

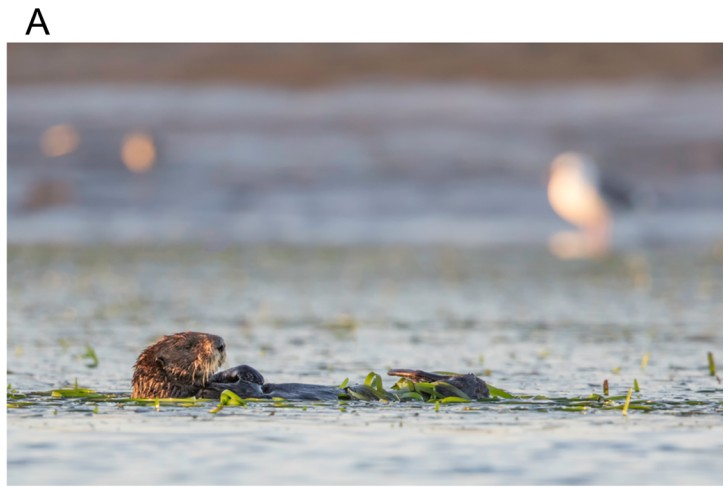

B

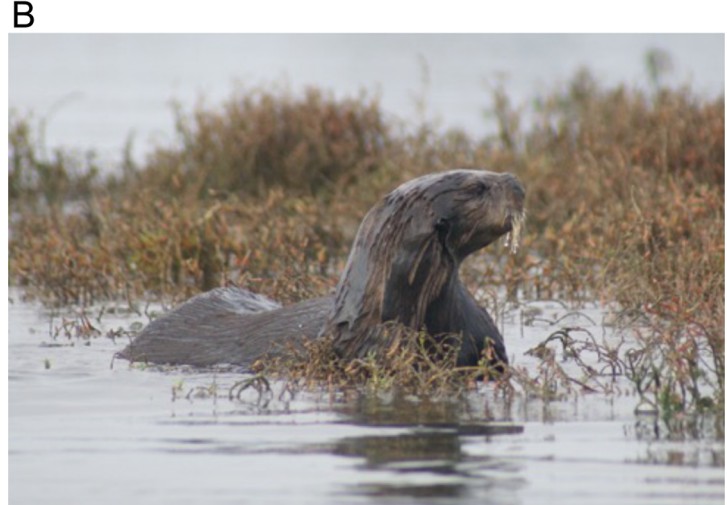

C

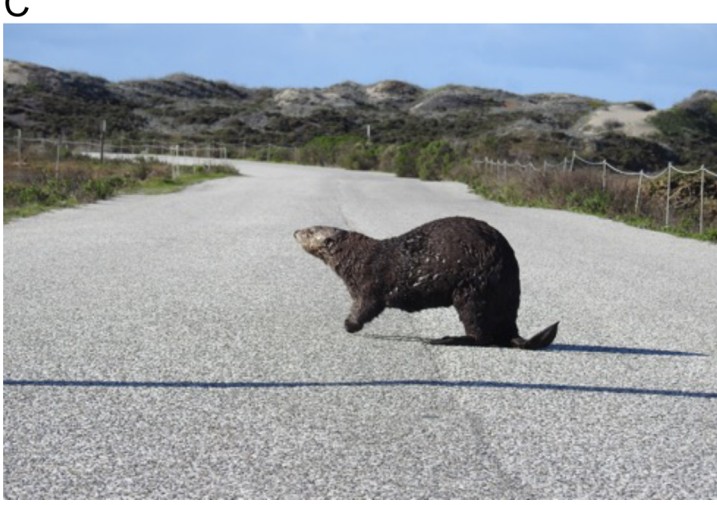

**Figure 4 Sea otters in estuarine and coastal habitats in Elkhorn Slough.** (A) Resting in an eelgrass bed (*Zostera marina*), (B) hauled out on pickleweed marsh (*Salicornia pacifica*), (C) a sea otter crossing the road between salt marshes. Photo credits: (A) Joseph A. Tomoleoni, (B) Ron Eby, (C) Natsuko Fujomoto.

(Elkhorn Slough and Morro Bay) have a significantly lower crab size and crab biomass for the two most abundant species of cancrid crab prey in California estuaries (*R. antennarium* and *Cancer productus*) than estuaries without sea otters (San Francisco Bay and Drakes Estero; Fig. 3). To maintain homeostasis, sea otters must consume, 25% of their body weight per day in invertebrate prey (*Costa & Kooyman, 1982*). As such, sea otters can have very strong effects on their prey (*Hughes et al., 2013*, *2016*; *Needles et al., 2015*), and have been documented to forage in a range of estuarine habitats, from seagrass beds to salt marshes (Fig. 4), exploiting a wide range of prey items, including crabs, clams, mussels, and worms (Table 2; Dataset S3).

While sea otters have diverse diets that include many non-commercial species, sea otter recolonization could affect commercial and recreational fisheries through competition or indirect effects. Sea otter predation can impact shellfish fisheries by reducing target species' abundance and size (*Garshelis & Garshelis, 1984*). In estuaries where oyster aquaculture is practiced, such as Tomales Bay or Morro Bay, it is conceivable that sea otters could affect oyster production, although no data are currently available to assess the likelihood or magnitude of such interactions. Whereas far fewer fisheries occur in California estuaries than on the open coast, estuaries provide important nursery habitat for some commercially exploited species. Sea otter effects on spawning adult Dungeness crab would likely be largely negative, while effects on Pacific herring (*Clupea pallasii*) would likely be positive through beneficial effects on eelgrass habitat. The magnitude of positive or negative effects on commercial or recreational fisheries is highly contingent on the particular ecological setting, the fisheries involved, the number of sea otters present, and other factors.

Despite the positive or negative effects of sea otters on fisheries, sea otter viewing has high recreational value and some people value their intrinsic presence and would view reintroduction favorably. Sea otters are immensely popular with the public, in aquaria worldwide and in their natural habitats. Visitors to Elkhorn Slough cite sea otter-watching as the number one reason to visit this coastal habitat (*Kildow & Pendleton, 2010*), and in situ cameras (www.elkhornslough.org/ottercam) showing real time footage of sea otters in the estuary receive tens of thousands of unique views each year. If sea otters become common along the shores of San Francisco Bay, millions of additional people could have the opportunity to observe and enjoy sea otters in habitat they historically occupied, and ecotourism opportunities and revenues could increase accordingly. Sea otters could also build support for ecosystem restoration of degraded seagrass and salt marsh habitats within estuaries (*Van Dyke & Wasson, 2005*; *Hughes et al., 2013*; *Wasson et al., 2017*). The linkage of sea otters to seagrass and salt marsh health could help motivate public support for conservation and restoration of the entire ecosystem.

While estuaries provide suitable habitat for sea otters, they also pose different threats to sea otters than the open coast because of their proximity to areas of intensive human use. Pollutant inputs from adjacent land use activities are often high, and pollutants can be concentrated in shallow estuarine waters with long residence times. For instance, extremely high dichlorodiphenyltrichloroethane (DDT) levels were measured in male sea otters from Moss Landing at the mouth of the Elkhorn Slough Estuary (*Jessup et al., 2010*), although these high contaminant loads have not been linked to lower rates of survival.
Sea otters at Elkhorn Slough are also affected by roads that disrupt the connectivity of historic estuarine wetlands (Fig. 4C). At two sites near the mouth of the Elkhorn Slough Estuary, sea otters routinely cross roads that bisect estuarine habitats; one sea otter was killed after being hit by a car in 2016, leading to efforts to slow traffic and enhance driver awareness in these areas. Depending on the size of the estuary, speeding boats and shipping traffic may also pose a risk to sea otters, either through direct strikes or through the potential for an oil or other chemical spill. Finally, repeated disturbance throughout the day caused by kayakers or eco-tourism boats may deprive sea otters of needed rest, deplete their energy reserves, and increase their vulnerability to other stressors (see https://www.seaottersavvy.org).

Elkhorn Slough provides a unique perspective when considering suitable estuarine habitats for sea otters outside their range. However, there are distinct challenges of human overlap in estuaries depending on the region and the individual estuary. San Francisco Bay, for example, has undergone dramatic physical and biological alterations as a result of human uses both past and present (*San Francisco Bay Subtidal Habitat Goals Report, 2010*). The Bay hosts four major shipping ports, two crude oil refineries, and a network of commuter ferries, in addition to many recreational boaters and wind surfers. Sea otters in San Francisco Bay would be exposed to the threat of injury from vessel traffic and could incur chronic fitness costs associated with repeated disturbance or displacement from preferred habitat. Heavy metal and organochlorine contaminants such as methylmercury, polychlorinated biphenyls, and DDT persist in high concentrations in San Francisco Bay (*SFEI, 2017*), and sea otters could be exposed to these chemical pollutants through consumption of contaminated invertebrate prey (*Jessup et al., 2010*). Large oil spills, although infrequent, are a potentially devastating threat to sea otters. Two oil refineries (Chevron and Phillips 66) operate on the Richmond waterfront of San Francisco Bay, receiving shipments of crude oil via tankers. San Francisco Bay experienced a major oil spill event in 2007 when the M/V *Cosco Busan* struck the Bay Bridge and spilled 53,569 gallons of fuel oil into the bay. Other threats associated with this urbanized estuary include an influx in bacterial disease associated with wastewater treatment plant discharges, entanglement in recreational or commercial fishing gear, exposure to harmful algal toxins, and destruction of foraging habitat and invertebrate prey by underwater dredging and sand mining operations.

If the reintroduction of sea otters to one or more estuaries were formally proposed, the engagement of the public and stakeholders could be used to fully explore the implications, including potential direct and indirect effects. Decision support tools and analyses that consider the full spectrum of trade-offs would be valuable to accompany and inform management decisions. Marine spatial planning, a process that enables decision-making around conservation trade-offs, is one such tool that can aid in the gathering and disseminating of information regarding the issues surrounding sea otter reintroduction into estuaries. This process could be especially useful when the needs and values of stakeholders become incompatible with conservation objectives. In Australia, *Atwood et al. (2015)* demonstrated the trade-offs between the recovery of shark predators, the abundance of herbivore bioturbators, and the seagrass system's ability to store carbon.

Similarly, in the northeast Pacific, key trade-offs include the recovery of listed species (sea otters), the abundance of their prey (including shellfish resources), and the ecosystem-level effects of sea otter reintroduction. Some of these ecological effects, while complex and difficult to monetize, may tip the balance between human values and top predator recovery that have historically pitted top predators against the humans competing with them for prey, and human harvest thereof. For example, the reintroduction of grey wolves into Yellowstone National Park has resulted in ecosystem benefits, such as increased woody vegetation and stream functioning, that could outweigh the costs (i.e., predation on livestock; *Ripple et al., 2014*). Comparison of estuaries occupied (Elkhorn Slough and Morro Bay) and unoccupied (San Francisco Bay and Drakes Estero) by sea otters offer the potential for valuable insight on socio-ecological effects of coastwide recovery. In order to realize the prospects of healthy coastlines, including the ecological restoration of estuarine systems such as seagrasses and salt marshes, differential human values and the ecological, cultural and economic trade-offs of sea otter recovery are crucial to consider.

## CONCLUSIONS

Our views of sea otter influences on coastal habitats and our understanding of the factors limiting their recovery have been grounded in open coast ecology. For the California population, this outcome was the result of a historical accident: specifically, that the recovering population spread from a remnant source population located on a remote stretch of rugged open coast. Recent sea otter studies from estuarine ecosystems have broadened conceptions of what we previously thought of as a kelp forest animal. Sea otters influence and are influenced by estuarine habitats in very different ways than their counterparts on the open coast. To understand their ecology and to support their recovery requires a paradigm shift in which we recognize and further explore the unique relationship of estuaries and sea otters.

## ACKNOWLEDGEMENTS

This publication is dedicated to the memory of our co-author, Susan Williams, whose leadership and creativity inspired this work. Sarah Codde and Gabriela Reyes (NPS) assisted with crab surveys in Drakes Estero. Jen Miller and Melissa Patten assisted KEB with crab surveys in San Francisco Bay. Paul Engel and Carola DeRooy (NPS) located historic sea otter remains from Native American middens near Drakes Estero. The findings and conclusions in this article are those of the authors and do not necessarily represent the views of the U.S. Fish and Wildlife Service or the US National Park Service. Any use of trade, firm, or product names is for descriptive purposes only and does not imply endorsement by the U.S. Government.

### Funding

Brent B. Hughes was funded through the David H. Smith Research Conservation Fellowship and Cedar Tree Foundation and the Rebecca and Steve Sooy Fellowship in

Marine Mammals. Kerstin Wasson was supported by a grant from NOAA's Office for Coastal Management to the Elkhorn Slough National Estuarine Research Reserve. Margot Hessing-Lewis and Erin Foster were supported by the Hakai Institute and Erin Foster was supported by an NSERC Vanier. Susan L. Williams was supported by the University of California's Agricultural Experimental Station. Michelle Staedler was supported by the Monterey Bay Aquarium. The funders had no role in study design, data collection and analysis, decision to publish, or preparation of the manuscript.

### Grant Disclosures

The following grant information was disclosed by the authors:
Research Conservation Fellowship and Cedar Tree Foundation and the Rebecca and Steve Sooy Fellowship in Marine Mammals.
NOAA's Office for Coastal Management to the Elkhorn Slough National Estuarine Research Reserve.
Hakai Institute.
University of California's Agricultural Experimental Station.

### Competing Interests

The authors declare that they have no competing interests.

### Author Contributions

- Brent B. Hughes conceived and designed the experiments, performed the experiments, analyzed the data, contributed reagents/materials/analysis tools, prepared figures and/or tables, authored or reviewed drafts of the paper, approved the final draft.
- Kerstin Wasson conceived and designed the experiments, performed the experiments, analyzed the data, contributed reagents/materials/analysis tools, authored or reviewed drafts of the paper, approved the final draft.
- M. Tim Tinker conceived and designed the experiments, performed the experiments, analyzed the data, contributed reagents/materials/analysis tools, prepared figures and/or tables, authored or reviewed drafts of the paper, approved the final draft.
- Susan L. Williams conceived and designed the experiments, authored or reviewed drafts of the paper.
- Lilian P. Carswell conceived and designed the experiments, authored or reviewed drafts of the paper, approved the final draft.
- Katharyn E. Boyer conceived and designed the experiments, performed the experiments, contributed reagents/materials/analysis tools, authored or reviewed drafts of the paper, approved the final draft.
- Michael W. Beck conceived and designed the experiments, authored or reviewed drafts of the paper, approved the final draft.
- Ron Eby conceived and designed the experiments, performed the experiments, authored or reviewed drafts of the paper, approved the final draft.

- Robert Scoles conceived and designed the experiments, performed the experiments, authored or reviewed drafts of the paper, approved the final draft.
- Michelle Staedler conceived and designed the experiments, authored or reviewed drafts of the paper, approved the final draft.
- Sarah Espinosa conceived and designed the experiments, performed the experiments, authored or reviewed drafts of the paper, approved the final draft.
- Margot Hessing-Lewis conceived and designed the experiments, authored or reviewed drafts of the paper, approved the final draft.
- Erin U. Foster conceived and designed the experiments, authored or reviewed drafts of the paper, approved the final draft.
- Kathryn M. Beheshti conceived and designed the experiments, performed the experiments, contributed reagents/materials/analysis tools, authored or reviewed drafts of the paper, approved the final draft.
- Tracy M. Grimes conceived and designed the experiments, performed the experiments, contributed reagents/materials/analysis tools, authored or reviewed drafts of the paper, approved the final draft.
- Benjamin H. Becker conceived and designed the experiments, contributed reagents/materials/analysis tools, authored or reviewed drafts of the paper, approved the final draft.
- Lisa Needles conceived and designed the experiments, performed the experiments, contributed reagents/materials/analysis tools, authored or reviewed drafts of the paper, approved the final draft.
- Joseph A. Tomoleoni conceived and designed the experiments, performed the experiments, authored or reviewed drafts of the paper, approved the final draft.
- Jane Rudebusch conceived and designed the experiments, performed the experiments, authored or reviewed drafts of the paper, approved the final draft.
- Ellen Hines conceived and designed the experiments, authored or reviewed drafts of the paper, approved the final draft.
- Brian R. Silliman conceived and designed the experiments, contributed reagents/materials/analysis tools, authored or reviewed drafts of the paper, approved the final draft.

## Field Study Permissions

The following information was supplied relating to field study approvals (i.e., approving body and any reference numbers):

The California Department of Fish and Wildlife granted approval to conduct field research (permit # SC-6563).

## Data Availability

All data and code are available in the Supplemental Files.

## Supplemental Information

Supplemental information for this article can be found online at http://dx.doi.org/10.7717/peerj.8100#supplemental-information.

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
