# Peer review of "Species recovery and recolonization of past habitats: lessons for science and conservation from sea otters in estuaries"

_PeerJ, doi:10.7717/peerj.8100_

## Round 0.1 · original submission · Major Revisions

While both reviewers are supportive of this publication, they have many suggestions. The communication of the manuscript needs a revision to include relevant details. In addition, the reviewers point out more substantial issues with the experimental design and assumptions which require consideration in the revision. Please address each point in full, as the reviewers have done an excellent and thorough review that will improve the manuscript.

Reviewer 1 ·

Basic reporting

Comments to Author:
Overall, I think this was a well-written manuscript on a worthy topic. The authors present a very interesting study on the potential for estuaries to support sea otter populations and aid in the recovery of southern sea otters. The authors created a habitat specific population model for sea otters in Elkhorn Slough and applied that model to estimate the number of sea otters that could be supported in San Francisco Bay, and estimated it could support 66,000 sea otters. They also found that prey populations in San Francisco Bay are sufficient to support a healthy sea otter population. I believe this study adds important information on sea otter recovery by studying sea otters in estuaries, for which fewer studies have been conducted when compared with studies of sea otters in outer coast habitats.
Major comments:
1. I don’t think that the historical importance of estuaries as sea otter habitat is best presented in a literature review in the Methods and Results. I think the results of this literature review are better suited for inclusion in the Introduction. The addition of this information to the Introduction will strengthen the background provided to give context for this study. Along these lines, on Lines 104-106, I suggest adding some additional discussion of the historical importance of estuarine habitats to sea otters, including beyond California, as sea otters have historically occupied estuaries in other areas, such as in Washington State.
There are a few areas where the phrasing could be improved for comprehension and/or clarity. I have listed these below, along with other minor comments/suggestions.
Minor comments/edits:
1. The abstract does not include any information on the results of the prey surveys that were conducted. I suggest adding a sentence or two to the abstract to summarize the results of this part of the study, given that the current abstract is 250 words of the allowed 500.
2. Line 61: Suggest changing “Recovering species often are limited to…” to “Recovering species are often limited to…”
3. Line 63: Suggest changing “accident” to “artifact of”
4. Line 66: Suggest changing “estuaries provide” to “estuaries can provide”
5. Line 79: Suggest changing “for successful restoration ecosystem functions and processes” to “for successful restoration of ecosystem functions and processes”
6. Line 87: Suggest changing “missing keystone predator” to “top predator whose decline has dramatically altered ecological relationships as a keystone predator” or something similar to more directly connect this statement about sea otters to the topic of the paragraph.
7. Line 96: Consider deleting “in the early 20th century” and “small” as they are repetitive with the 1914 reporting date and 50 individuals
8. Line 98/99: Suggest changing “As of 2017, this population has grown to over 3,000 individuals and has expanded to the north and south along a range that extends more than 400 km (USGS, 2018).” To “As of 2017, this population has grown to over 3,000 individuals and has expanded its geographic range to include more than 400 km of coastline (USGS, 2018).”
9. Line 101/102: Estes and Palmisano 1974 and Simenstad et al. 1978 are cited for “foundational studies on the ecology of southern sea otters” that have focused on kelp habitat, but these studies are on Northern sea otters in the Aleutians. I suggest either deleting these citations and replacing them with relevant citations specific to southern sea otters, or broadening the statement to say something along the lines of “because many remnant recovering populations occurred in kelp forests along open coastline, foundational studies of the ecology of sea otters have focused on this habitat”.
10. Line 105: I suggest deleting the Laidre et al 2001 citation. I think it is potentially misleading to cite Laidre et al 2001 to support the statement that “southern sea otters historically occupied estuaries as well as the open coast” as the use of the word “historical” in this context is assumed to mean pre-fur trade, while the citation is well after fur trade and does not discuss estuaries historically, but in the mid-1990s.
11. Lines 109-111: I suggest adding relevant citations here
12. Lines 109-111: This sentence appears to be out of place, as it refers to “these open-coast studies” when the beginning of the paragraph describes sea otter use of estuaries. I suggest deleting “these” from this sentence.
13. Lines 111/113: Both “open-coast” and “open coast” are used throughout. I think “open coast” is correct. Either way, it should be consistent throughout.
14. Line 118: Suggest adding citations (Hessing-Lewis et al., 2017; Hughes et al., 2016, 2013) after “recent and ongoing research”
15. Line 122: Add “to” between “sea otters” and “better”
16. Line 122-124: I am not sure that this work actually accomplishes the goal of using “the recovery of southern sea otters to better understand how a recovering species interacts with historically occupied ecosystems..”. I suggest removing this sentence.
17. Line 143: I think the asterisk should be moved to directly after “embayment” instead of after “embayment or”
18. Line 154/155: Suggest adding a citation to support the assertion that “it is likely that other estuaries (including San Francisco Bay) will eventually be colonized”. Has any population modeling predicted this?
19. Line 168: Consider adding “growth” between “stochastic logistic” and “model”
20. Line 172: Suggest removing “hidden or”, as “hidden” and “latent” are synonymous. If the goal was to provide a definition of the term “latent”, I would suggest saying “…true abundance, Nt, is an unobserved (latent) variable…” instead.
21. Line 176: Should equation referenced be 0.1 instead of 1? Similar issue throughout if so.
22. Line 177/178: I am confused by the inclusion of both  and p, as only p is included in Equation 0.1. I think it would be more clear and correct to say “…N(0, p) represents a random normal variable with mean 0 and standard deviation p, where p represents process error or environmental stochasticity (a fitted parameter).”
23. Line 189: Consider adding a citation for JAGS and R.
24. Line 190: Consider adding a comma after K, as it sounds like K and N1 are variance parameters as written.
25. Line 191: I think it is misleading to say that this is a weakly informed prior. I would remove “weakly” here.
26. Line 205-208: The four habitat types should be listed explicitly here, or earlier near line 180.
27. Line 206: Would like to see a citation here for degradation of San Francisco Bay habitats.
28. Line 209: capitalize bay
29. Line 212: The use of the phrase “somewhat lower densities” seems at odds with the decision to assume that offshore densities would be ~10% of nearshore densities. Perhaps this could be addressed by explicitly stating average distances from shore or percent of time spent offshore in embayments in AK or other areas instead of saying “somewhat lower densities”. These citations could be useful here (although are foraging specific):
Bodkin, J. L., G. G. Esslinger and D. H. Monson. 2004. Foraging depths of sea otters and implications to coastal marine communities. Marine Mammal Science 20:305–321.
Laidre, K. L., R. J. Jameson, E. Gurarie, S. J. Jeffries and H. Allen. 2009. Spatial habitat use patterns of sea otters in coastal Washington. Journal of Mammalogy 90:906–917.
30. Line 220: Suggest adding “derived from Elkhorn Slough” after “habitat-specific equilibrium densities”.
31. Line 220-222: This seems like it should be in the results instead of the methods. I think “of 6607 otters (Table 1)” should be removed, and the methods should be discussed generally. I also suggest deleting reference to Fig. 2A-B, as these are results as well.
32. Line 223: It is unclear to me as written if r and p are the parameters estimated from the Elkhorn Slough data, or if they are estimated for San Francisco Bay, although I believe they are the parameters estimated from the Slough data. I suggest explicitly stating this.
33. Line 236-238: The current phrasing of this section makes comprehension difficult. I suggest making it clear on line 237 where prey species lists were “assembled” from by moving the information about the prey species lists on lines 240-242 earlier. I suggest then combining the information about crab sampling and sea otter prey surveys into one sentence.
34. Line 251: Define CPUE here.
35. Line 259: Does “conspicuous” mean they are larger than other crabs? Or present in the Slough at higher abundances? Suggest adding more detail here.
36. Lines 260/261: CPUE definition can be deleted here if it is defined when first used on line 251.
37. Line 261: Add “s” to “linear mixed model”, or change to “using a linear mixed model”
38. Line 272: Change citation for “Seachore” to “Seashore”
39. Line 287: Perhaps “continued to be published” should be changed to “continue to be published”
40. Line 327: Suggest specifying that the range referenced is for Southern sea otters – as written, this statement could be misinterpreted to mean all sea otters, not just Southern sea otters.
41. Line 330-332: This sentence seems more appropriate for the Discussion rather than the Results
42. Line 338-339: This sentence also seems more appropriate for the Discussion rather than the Results
43. Line 338: Here both SF Bay and Drakes Estero are listed as “focal sites”, but above on line 331 SF Bay is listed as the focal recovery site. On Line 331, I suggest saying “one of the focal recovery sites”
44. Line 339: Suggest adding “to support a sea otter population” after “supply of crab prey”
45. Lines 342-347: The current phrasing of this section makes comprehension difficult. How can crabs in Drakes Estero be of similar size to crabs in Elkhorn Slough pre-sea otter, and crab sizes in SF Bay also intermediate between Drakes Estero, Morro Bay, and Elkhorn Slough (pre and post sea otter)?
46. Line 343: Suggest replacing “greater” with “larger”, and adding “crabs from” between “larger than” and “sites”
47. Line 368: Is “topographically” the appropriate term here? I think it is if you are referring to the shallow water in estuaries, but I think a broader term would be better suited here that describes the habitat complexity that estuaries provide.
48. Line 382: It is unclear what is meant by “natural experiment” in this context, although I have an idea based on the use of recolonization as a natural experiment in other sea otter research. Do you mean sea otter recolonization of the Slough as a natural experiment to test if sea otters can thrive in an estuary? I suggest clarifying this.
49. Lines 415-418: The current phrasing of this section makes comprehension difficult. Suggest changing “When comparing the coverage of the two dominant habitat-forming species (estuaries with eelgrass) and the outer coast (with kelp, e.g., Macrocystis pyrifera) we conservatively estimated there to be 1750 ha of eelgrass habitat within the projected range, and most of the habitat is from the northern side of the otter range (see Bernstein et al. 2011).” To: “When comparing the coverage of the two dominant habitat-forming species within the sea otter range – eelgrass in estuaries and kelp (e.g., Macrocystis pyrifera) on the outer coast – we conservatively estimated there to be 1750 ha of eelgrass habitat within the projected range, and most of the habitat is from the northern side of the otter range (see Bernstein et al. 2011).”

In addition, it is unclear what is meant by “projected range” – do you mean the 100km to the north or south of the current range? Finally, this sentence does not actually compare the coverage of kelp and eel grass, despite saying “When comparing the coverage of the two dominant habitat-forming species…”. I suggest adding the ha of kelp habitat here, although I do see these discussed later in the paragraph.
50. Line 431: Suggest changing “At present” to “As of 20XY” (applicable year).
51. Line 433: Should it be “has a relatively pristine watershed” instead of “had”?
52. Line 455-458: Suggest changing from “Results showing positive effects of sea otters in estuaries has regional conservation implications...” To: “Results showing positive effects of sea otters in estuaries have regional conservation implications...”
53. Line 458-460: This is not a complete sentence. Suggest changing “Severe recent losses of eelgrass to the south of Elkhorn Slough in Morro Bay (> 95% since 2007; (MBNEP, 2014)), with the exception of areas near the mouth where sea otters permanently reside.” To: “Recently, severe losses of eelgrass to the south of Elkhorn Slough in Morro Bay (> 95% since 2007; (MBNEP, 2014)) have been observed, with the exception of areas near the mouth where sea otters permanently reside.”
54. Line 475: Suggest adding applicable citations here.
55. Line 482 and 485: Capitalize estuary
56. Line 491: Is there applicable work you can site here on sea otter disturbance? Like work by Sea Otter Savvy?
57. Line 515-516: Suggest adding applicable citations here
58. Line 534: Suggest changing “where they have been documented to hunt” to “and have been documented to forage in a range of estuarine habitats…”
59. Line 540: I have never heard of sea otters eating oysters. As such, I think it is important to mention (and cite applicable literature) if sea otters have ever been observed eating oysters anywhere throughout their range, and if they have not been observed eating oysters, say so.
60. Lines 563-581: This section seems repetitive with Lines 477-491. Consider combining and consolidating.
61. Line 588: Re-introduction is hyphenated here but not elsewhere. Suggest not hyphenating and being consistent throughout.
62. Line 599: Unclear what is meant by “comparison between estuaries…”. Suggest changing to “Comparison of estuaries occupied (Elkhorn Slough and Morro Bay) and unoccupied (San Francisco Bay and Drakes Estero) by sea otters offer the potential for valuable insight on socio-ecological effects of coastwide recovery”
63. Line 614: The current phrasing of this section makes comprehension difficult. Do you mean “Engaging in restoration of both species and habitats?”, or “both habitats” as in eelgrass and kelp? In addition, “implementation requires an active, adaptive approach” is vague. Implementation of what?
64. Lines 625: should read “assisted with crab surveys in Drakes Estero”
65. Line 848: The figure legend refers to “protected coastal habitats” but the manuscript uses estuary throughout. I suggest being consistent with the use of estuary, and changing this in the figure legend.
66. Line 849: Suggest changing the caption to better describe blue points to say: “(A) Range of southern sea otters in California estuaries as of 2016. Red points indicate protected coastal habitats where permanent populations currently exist. Blue and black points indicate estuaries within 100 km of the current range (USGS, 2018), and blue and red sites are focal sites used in this study…” In addition, the red line is not explained in the figure caption, and I suggest doing so for clarity.
67. Line 858: The caption states “equilibrium abundances” but the figure legend says “density at K”. Which one is correct?
68. Population model code does not appear to have been attached.

Experimental design

Major comments
1. The Introduction needs more discussion of the knowledge gap being filled by this research, in addition to that provided on Lines 117-121. Along those lines, one way to provide justification and context for this work is to discuss the current status of southern sea otters in the Introduction. I suggest moving Lines 464-472 into the Introduction (or stating something similar in the Introduction), as it helps to clearly articulate the knowledge gap and impetus for this work. Similarly, in order to provide additional context, I suggest discussing the results from Silliman et al. 2018 (i.e. that many animals, rather than occupying these seemingly unique habitats for the first time, are in fact reoccupying historical habitat). I could see this fitting in well around Line 100.
2. The methods are currently not written with sufficient detail and information to replicate. The sizes of crabs in the 1970s are not brought up in the Methods, and are first mentioned on Line 345. I suggest making these time periods more explicit in the Methods. As the methods are currently written, it appears that current prey species composition and crab biomass and size were compared between sites within and outside the current sea otter range. This would be an issue, as sea otter presence over time at sites within the current range would likely change the biomass and size of crabs available, hence mentioning the pre-sea otter colonization sizes and biomass is important.
3. Similarly, there is not enough detail included in the crab capture methods on Lines 244-256. What effort went into surveying crabs? How many days were pots put out? In what seasons? At different points in the tidal cycle?
4. It is not clear as written on Lines 238-243 and Line 332 if the prey in the “prey species lists” have been observed as sea otter prey items, or if these benthic prey were identified as “possible” sea otter prey items during benthic surveys. Have all the prey types listed been observed being consumed by a sea otter? I think sea otter diet data and benthic invertebrate surveys should go into making this list. In Table 2, oysters are listed, which are not a known sea otter prey item to my knowledge, and the literature cited are for benthic surveys alone. I think Supplementary Table 2 is much more informative and clear in this regard.
5. I am not convinced that assessing the biomass and size of two crabs and the presence/absence of other prey in each estuary is a justifiable metric of sea otter prey availability and therefore potential for expansion into an area. I think this choice needs further discussion and justification on Lines 238-240.

While Hughes et al. 2013 and Garshelis and Garshelis 1984 are cited, I do not think that these citations support the assertion that crabs “are considered to be an essential prey item for successful sea otter recolonization”, even though Hughes et al. 2013 did find that the majority of the diet in Elkhorn Slough was made up of crabs in the last 10 years. I think the jump between crabs being the majority of the diet in some currently occupied areas and therefore being “an essential prey item for successful sea otter recolonization” is too big. While I agree that crabs are an important prey item for sea otters in many areas, I think this assertion needs further support from the literature in addition to Garsehlis and Garshelis 1984 and Hughes et al. 2013. What about areas that sea otters have recolonized where bivalves are the primary prey item? How can crabs be an “essential prey item for successful sea otter recolonization” in areas where they are rarely eaten?

Minor comments/edits
1. If you decide to leave the literature review in the Methods/Results, I suggest a visual of the divisions used for categorizing the literature on Line 143-148, like a flowchart.

Validity of the findings

Minor comments/edits
1. Line 420: It is unclear how the 31% is calculated. If eelgrass habitat (1750 ha) and kelp habitat (~5700 ha) is what constitutes sea otter recovery habitat for this calculation, then eelgrass habitat would make up approximately 23% of available sea otter habitat.
2. The results of the literature review for northern and southern sea otters in different ecosystem types and habitats mostly focus on southern sea otters, does not discuss different habitat types within the ecosystems, and only briefly discuss northern sea otters. If you choose to keep the literature review as it’s own section of the Methods and Results, the results of the literature review should be discussed in greater detail.

Additional comments

Major comments
1. The authors fail to mention that surrogate reared pups have been and are currently released into Elkhorn Slough. It is misleading to leave this out and not address how this has or has not influenced the population in the Slough, especially given that surrogate reared animals account for the majority of the population in the Slough. How does the population model account for these surrogate reared pups? Are they included or excluded? Similarly, the reintroduction of sea otters to other areas of historical occupation (like SF Bay) through the surrogate program has very much been a part of the conversation in recent years, especially with the Slough appearing to be at or near K, but the potential reintroduction of surrogate reared pups to SF Bay is not mentioned in the Introduction. This could provide additional justification and context for this work by exploring the potential of other estuaries to support sea otters.
2. If the Introduction is edited to include additional discussion of potential sea otter reintroduction to SF Bay, I suggest also bringing up historical reintroductions to SE AK, BC and WA around Lines 91-94. I think this will be important to mention, given that sea otter reintroductions have been instrumental in sea otter recovery post fur trade (pages 44-59 of Sea Otter Conservation book, 2015).
3. Line 539-540: I think sea otter/fisheries interactions warrant further discussion, and additional citations beyond Garhselis and Garshelis (1984). I suggest a brief but more comprehensive summary of the perceived conflict (and in some cases documented impacts) of sea otters on fishery species, and citing the work below.
Fanshawe, S., G. R. VanBlaricom, and A. A. Shelly. 2003. Restored top carnivores as detriments to the performance of marine protected areas intended for fishery sustainability: a case study with red abalones and sea otters. Conservation Biology 17:273–283.
Larson, S. D., Z. N. Hoyt, G. L. Eckert, and V. A. Gill. 2013. Impacts of sea otter (Enhydra lutris) predation on commercially important sea cucumbers (Parastichopus californicus) in southeast Alaska. Canadian Journal of Fisheries and Aquatic Sciences 70:1498–1507.
Hoyt, Z. 2015. Resource Competition, Space Use and Forage Ecology of Sea Otters in Southern Southeast Alaska. 152 pp.
Minor comments:
1. Line 125-127: Food for thought: The amount of research on sea otters in different coastal ecosystems (open coast vs. estuaries) is also determined by where estuary habitat is available for them to occupy/reoccupy and how much of it is available, in addition to where sea otters have been reintroduced/where the extant population remained, and/or research priorities. It would be interesting to also include a review of the literature that synthesizes where sea otters have estuary habitat available and have or have not reoccupied those areas. It seems to me that there would be much less habitat in estuaries than open coast habitat, but it would be interesting to see the numbers on that.
2. Line 362-364: Food for thought: This “narrow configuration” also seems to be the case in Washington, where sea otters are limited to a predominately one-dimensional habitat with range expansion only occurring at the range edges. However, as stated above, northern sea otter populations (including in Washington) are growing faster that southern sea otters in California. How do you account for this discrepancy, despite similar shoreline complexity/habitat configuration?
3. Lines 398-399: I suggest further discussion of sea otter haul out activity elsewhere to provide a broader context for this point. Is this haul out frequency unique to the Slough due to the availability of salt marsh habitat? What is the haul out frequency of sea otters elsewhere in their range (not just on the outer coast of CA)?
4. Line 454: After the discussion of the sea otter trophic cascade in eelgrass habitats, I am left wondering how this connects to sea otter recolonization, especially of San Francisco Bay. I suggest explicitly mentioning the projected benefits of sea otter recolonization into SF Bay, with potential for mediation of eutrophication and expansion of eelgrass beds, in the last sentence of this paragraph (even though they are discussed further in later paragraphs of the discussion).
5. Lines 251-256: Suggest specifying that green crab is not included in Oftedal et al 2007, thus necessitating the use of a function from another crab.
6. Line 486: Suggest adding year that sea otter was hit

·

Basic reporting

I think it reasonably good. The narrative is very long and comes across more as a position of advocacy then purely objective analysis. That said, I agree with many of the arguments and believe others close to the issues will as well.

Experimental design

NA

Validity of the findings

The analyses involve a fair number of unevaluated assumptions but otherwise I find the results to be fair and reasonable

Additional comments

Review of Hughes et al. for PeerJ

I should begin with the disclaimer that I am not broadly familiar with PeerJ and thus am not well suited to judge this ms. because of my general ignorance of the journal’s standards of rigor, scope, length, etc.

Hughes et al. provide an assessment of California’s estuaries as future habitat for sea otters; review the ecological consequences of repatriating sea otters to estuarine systems; and engage in a rather length dialogue (via the Discussion), mainly as proponents, for the importance of estuaries in the future conservation and management of southern sea otters. Some of the narrative is redundant with things that have already been written but the analyses, especially those having to do with sea otter population growth and abundance potential, are rigorous, novel, and very important to any future dialogue about sea otter management in California. This in my view is the strength. Although the ms. is a long read, I found it to be factual, reasonably objective, and worthy of publication in the scientific literature. The Discussion goes on and on to discuss the sundry issues concerning the repatriation of sea otters into estuarine habitat and mostly to advocate for this course of action. But in fairness, they also mention and discuss the costs and downsides of doing this, albeit seemingly to a lesser degree.

I don’t have major concerns or suggestions, owing largely from the disclaimer above. There is nothing I can see that is technically wrong with the ms. and it does provide information that will be of interest and use to scientists, managers, and those involved with policy.

Minor suggestions or points in need to further discussion and resolution

115. high production and ample food,

122. sea otters to better understand . . .

The multiple usage of “to better understand” in this paragraph is awkward. I would revise accordingly.

Equation 0.1. This is indeed a widely known model for density-dependent population change. I have two concerns about its use here. One is the assumption of linear density dependence (i.e., as a function of N/K). Most of the empirical analyses and data I’m familiar with for large mammals seem to indicate that per capita density dependence is not linear but instead increases more sharply near K. This is not a fatal flaw but rather something to consider in the revision. My other concern is over the assumption of error with mean = 0, which implies that counts are unbiased. This is almost never true for wildlife surveys as there is always some probability of an animal being missed. There are methods for estimating the probability of sighting but they have not been applied to these data, at least so far as I am aware.

I am not competent to evaluate the statistical methods described between lines 185 and 202. I think it important for someone to check through this who is.

238. Despite Garshelis 1984 and the various observational data of sea otter diet, I’m not entirely comfortable with the claim that crabs are essential for sea otter colonization. I see no logical reason why that should be. Instead, it may simply be that most or all estuaries support decapods and thus sea otters consume decapods when and wherever they have redolonized estuarine habitats. I simply can’t understand why other groups of prey would not be suitable prey in this process, even if crabs were entirely absent. Unless the authors provide better supporting evidence, I think they need to rethink and possible reargue this element of the ms.

317. “Future work to improve precision of these parameter estimates will reduce the uncertainty in model projections.” This statement strikes me as being almost patently obvious on the one hand and probably untrue on the other. That is, I’m not convinced that the major uncertainties and surprises in population trends are the result of parameter imprecision rather errors or deficiencies in model structure.

---

## Round 0.2 · Minor Revisions

I appreciate the effort taken in the revision, but I found a few issues that I think will improve your manuscript.

Abstract: I would suggest adding a link (logic) between the second to last and last sentence of the abstract.

Text: The text has a lot of overlap in what is discussed and presented. Please re-read the introduction and discussion and reduce redundancies. For instance, the benefits of estuaries for otters are described in the introduction (starting at Line 139) and in the discussion (starting at Line 447).

Figures: The figures look they are from ggplot and have not been modified to suit publication. For instance, some of the text (axes labels are really small) and it is arguable if the gridded background is required. I would suggest that you take another look at the figures and modify their visual appeal for publication, including ensuring the text size of the fonts are within a reasonable (i.e., the difference in size between the axis tick labels, axis label and panel label is large).

---

## Round 0.3 · accepted · Accept

Apologies for the delay and congratulations on your manuscript.